Subject Area:
developmental biology

Keywords:
odd paired, Zic genes, crustacean cardioactive peptide, bursicon, ecdysis, *Drosophila*

Authors for correspondence:
Eléanor Simon
e-mail: esimon@cbm.csic.es
Isabel Guerrero
e-mail: iguerrero@cbm.csic.es

# *Drosophila* Zic family member odd-paired is needed for adult post-ecdysis maturation

Eléanor Simon, Sergio Fernández de la Puebla and Isabel Guerrero

Centro de Biología Molecular 'Severo Ochoa' (CSIC-UAM), Nicolás Cabrera 1, Universidad Autónoma de Madrid, Cantoblanco, 28049 Madrid, Spain

ES, 0000-0002-9799-2418; IG, 0000-0001-6761-1218

Specific neuropeptides regulate in arthropods the shedding of the old cuticle (ecdysis) followed by maturation of the new cuticle. In *Drosophila melanogaster*, the last ecdysis occurs at eclosion from the pupal case, with a post-eclosion behavioural sequence that leads to wing extension, cuticle stretching and tanning. These events are highly stereotyped and are controlled by a subset of crustacean cardioactive peptide (CCAP) neurons through the expression of the neuropeptide Bursicon (Burs). We have studied the role of the transcription factor Odd-paired (Opa) during the post-eclosion period. We report that *opa* is expressed in the CCAP neurons of the central nervous system during various steps of the ecdysis process and in peripheral CCAP neurons innerving the larval muscles involved in adult ecdysis. We show that its downregulation alters Burs expression in the CCAP neurons. Ectopic expression of Opa, or the vertebrate homologue Zic2, in the CCAP neurons also affects Burs expression, indicating an evolutionary functional conservation. Finally, our results show that, independently of its role in Burs regulation, Opa prevents death of CCAP neurons during larval development.

# 1. Introduction

Ecdysis is a physiological process that takes place in insects during transition stages or moults permitting an increase in body size or/and changes in morphology. After the final ecdysis or eclosion, the adult emerges from its cocoon or pupal case. To this end, a highly stereotyped programme takes place including wing expansion, thorax extension and hardening and pigmentation of the cuticle (reviewed in [1]). The crustacean cardioactive peptide (CCAP) neurons, identified in crustaceans and insects, are important regulators of the post-ecdysis process. CCAP neurons are neurosecretory cells that synthesize the neuropeptide Bursicon (Burs) [2]. Burs is released to the haemolymph at the onset of metamorphosis to bind to its receptor Rickets (Rk), the G protein-coupled receptor LGR2, resulting in a transient increase of cyclic AMP in target tissues [3,4]. Precise ablation of CCAP neurons by ectopic expression of the proapoptotic gene *reaper* (*rpr*) leads to defects in the post-ecdysis sequence [5]. Similar defects are observed in mutant alleles of either *Burs* [6] or *rk* [7,8]. CCAP neurons are highly specialized peptidergic cells that, besides Burs, express two other neuropeptides: the CCAP and the myoinhibitory peptides (MIP, also known as AstB) that are also involved in insect ecdysis (reviewed in [9,10]), although their specific implication in *Drosophila* ecdysis is unclear.

In *Drosophila*, the CCAP neurons are organized into a well-defined segmental pattern. In the ventral nerve cord (VNC), each hemisegment contains one CCAP interneuron and, depending of the hemisegment, one CCAP motoneuron nearby. CCAP inter- and motoneurons are not siblings but both belong to the lineage of neuroblast (NB) 3–5 [11]. CCAP neurons are generated early in development, during the first two rounds of division of NB 3–5. The majority of CCAP neurons have their hormonal differentiation in the embryonic stage 17 [11] except a subset of CCAP cells localized in the posterior VNC that differentiate much later, at the entrance of pupal stage [12]. Burs starts to be released along the CCAP axons prior to metamorphosis and its expression completely disappears in 24 h-old adult flies

royalsocietypublishing.org/journal/rsob    *Open Biol.* **9**: 190245

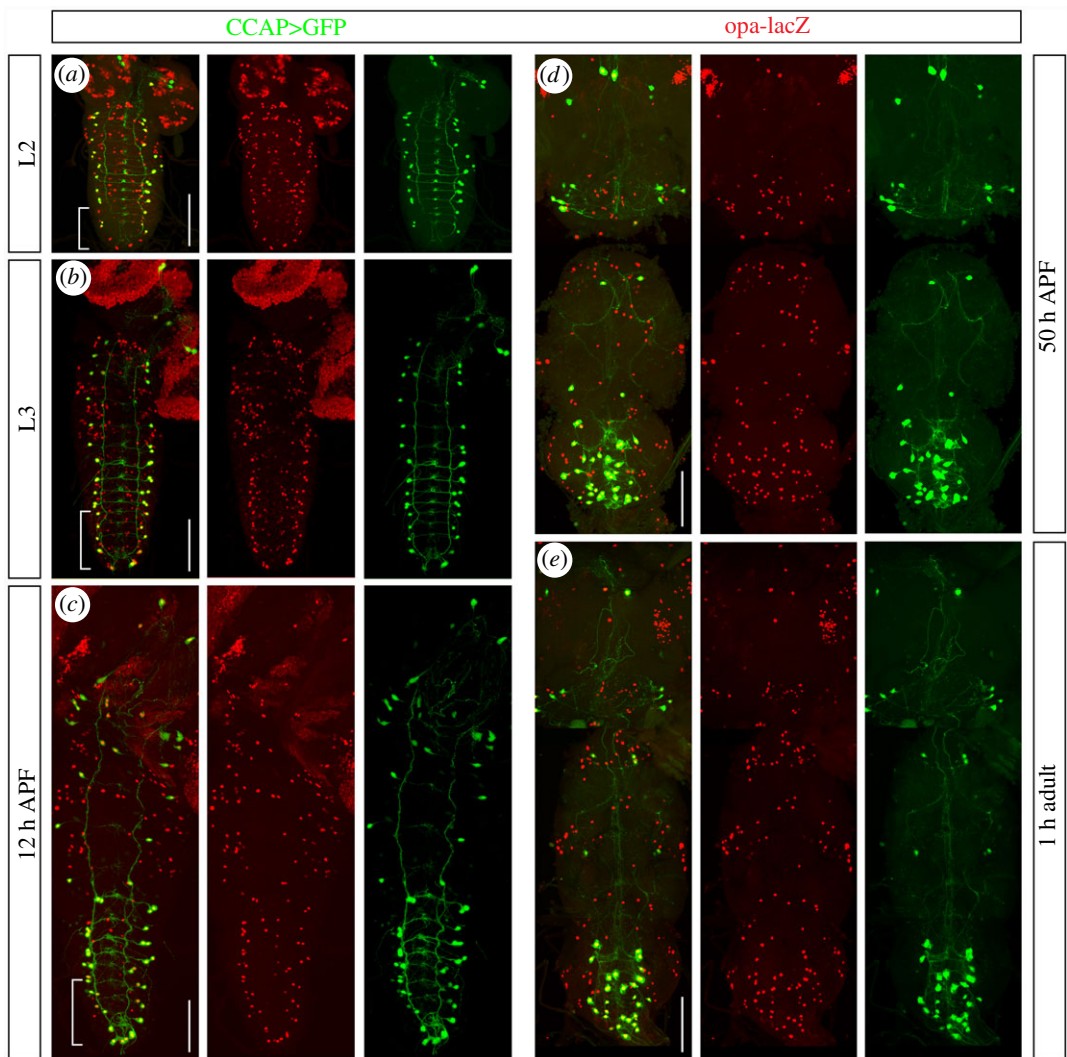

**Figure 1.** *opa* expression in the CCAP neurons in larval stages, pupae and adult. (*a*) L2, (*b*) L3, (*c*) 12 h APF, (*d*) 50 h APF and (*e*) 1 h adult CNS showing that all the CCAP neurons express *opa-lacZ*. Pictures correspond to maximum projections of stacks of the entire CNS along the dorsoventral axis. Note that the expression of *opa* (*a*; bracket) starts before to the expression of *CCAP-Gal4 UAS-GFP* in the most posterior CCAP cells (*b,c*; bracket). Scale bar, 100 µm. Anterior is towards the top.

[13]. The CCAP neurons get progressively eliminated by apoptosis during the first 4 days after eclosion [13,14].

In this study, we report an unexpected role of the *odd-paired* (*opa*) gene in the CCAP neurons. *opa* encodes a transcription factor which belongs to the conserved Zinc finger in the Cerebellum (Zic) family. Being a pair-rule gene, *opa* is required in *Drosophila* for the parasegmental subdivision of the embryo [15,16]. *opa* is also required for the morphogenesis of embryonic midgut constrictions [17] and the adult head [18]. It is also expressed in the glial cells of the optic lobe [19–21], the peripheral glia [22] and in intermediate neural progenitors where Opa regulates temporal patterning [23]. We show here that *opa* is expressed in all CCAP cells of the central and peripheral nervous systems and that either *opa* loss of function or its misexpression disturbs the post-ecdysis maturation by altering *Burs* expression. We also show that Opa prevents death of the CCAP neurons during larval development.

## 2. Results

### 2.1. Opa is a marker of the CCAP neurons in the CNS

We have studied the role of the transcription factor Opa during the post-eclosion period of *Drosophila*. We found that *opa* is expressed in all the CCAP cells of the central nervous system (CNS), motoneurons and interneurons, from the embryonic stage 12 (electronic supplementary material, figure S1) to the larval (figure 1*a,b*), pupal (figure 1*c,d*) and adult stages (figure 1*e*), indicating that *opa* is expressed in the CCAP neurons throughout their lifetime. *opa* expression is relatively limited, but not exclusive, to the CCAP neurons. In the dorsal part of the VNC, *opa* expression is restricted to the CCAP cells (figure 2*a*); the rest of the neurons that are not CCAP, but express *opa*, are found in the ventralmost part of the VNC. We have identified some of these cells as VAs neurons that express the neuropeptide myomodulin [24] and are marked by the *386Y-Gal4* driver (figure 2*b*) [25].

In *Drosophila*, most of the CCAP neurons differentiate as hormone producers during embryogenesis. A subset of 12 posterior CCAP neurons starts expressing the neuropeptide prior to metamorphosis under the control of ecdysone signalling. There is no consensus about the role of these neurons: while a study proposed they are necessary and sufficient for pupal ecdysis [12], another one indicated they are not [26]. Since no specific markers were previously described for the CCAP, it had been proposed, based on lineage tracing analysis, that these neurons are generated during embryogenesis but achieve their terminal differentiation at pupariation [12]. We can now assert these findings using *opa* as a reliable marker for all CCAP neurons.

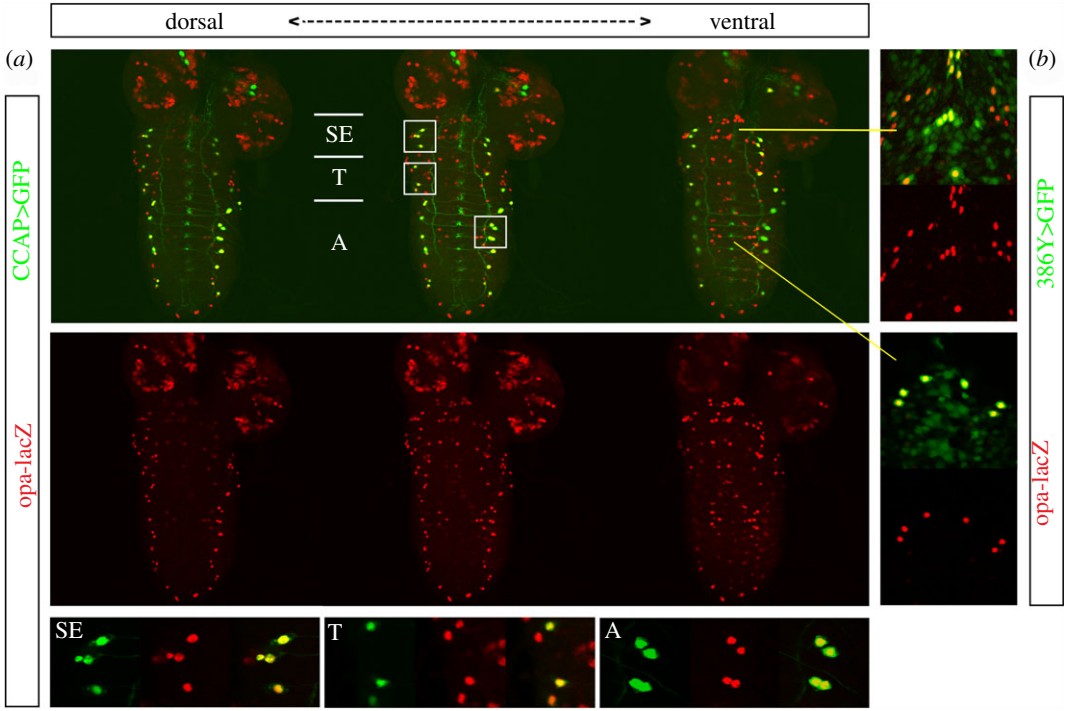

**Figure 2.** Larval expression of *opa* along the dorsoventral axis of the CNS. (*a*) In the dorsal VNC, *opa* is expressed in a segmental manner overlapping with the expression in the CCAP neurons in the thoracic and abdominal neuromeres. In the ventral VNC, *opa* is expressed in the CCAP neurons of the suboesophagic domain. It is also expressed in more ventral neurons of the suboesophagic region and in some ventral abdominal neurons. Images correspond to frames of the maximal projection presented in figure 1*a*. A zoom of the regions marked by a white square is shown in the lower panel. SE, suboesophagic segments; T, thoracic segments; A, abdominal segments. Anterior is towards the top. (*b*) In the ventral side of the VNC, *opa* is also expressed in the VAs neurons, marked by the *386Y-Gal4* driver in the ventral suboesophagic region (top inserts) and in the ventral abdominal segments (bottom insets). The yellow lines indicate the position of these neurons in (*a*).

Opa early expression allows tracking CCAP cells through development and confirms the prediction that those late terminal differentiated CCAP neurons are indeed generated during early development (see brackets in figure 1*a*–*c*).

## 2.2. Downregulation of *opa* expression in the CCAP neurons alters post-ecdysis maturation by controlling Burs expression and CCAP survival

We have identified *odd-skipped* (*odd*) as an Opa target in the eye-antenna imaginal disc. When mutant clones for a null allele *opa* (*opa^7^*) are induced in the antenna imaginal disc the expression of *odd* disappeared (electronic supplementary material, figure S2C, arrow and S2A,B for *opa* and *odd* wild-type expressions). We then use *odd* expression as read out of Opa loss of function to test if Opa RNAi expression gives the same result in clones induced in the eye-antennal disc as *opa^7^* mutant clones. *UAS-Dcr2* was coexpressed to increase the efficiency of the *UAS-RNAi-opa^II^* and we observed that *odd* expression disappears (electronic supplementary material, figure S2*d*, arrow). Once we were sure of the efficiency of *UAS-RNAi-opa^II^* with *UAS-Dcr-2*, we proceeded to knockdown *opa* function specifically in the CCAP neurons by expressing *RNAi-opa^II^* in these cells.

After emerging from the pupal case, flies expand their wings within a period of 30 min. The cuticle of the thorax also expands until the two posterior scutellar bristles, which cross each other in the pharate adult, become parallel. The sclerotization and darkening of the cuticle take place during the first 3 h after eclosion (reviewed in [1]). All these characteristic changes can be seen in the 3 h-old *CCAP-Gal4*

*UAS-Dcr-2* control flies (figure 3*a*). In order to figure out the role of *opa* during post-eclosion, we expressed ectopically the *UAS-RNAi-opa^II^* and *UAS-Dcr-2* in the CCAP cells through development. In the 3 h-old flies, we noted a lack of wing and thorax expansion, evidenced by the misorientation of the posterior scutellar bristles. In addition, the cuticle fails to harden and the pigmentation is absent (figure 3*b*). No lethality is associated with this genotype since all flies reach adulthood and pharate adults do not have apparent patterning defects. In order to explore when *opa* is required we temporally controlled *RNAi-opa^II^* expression using Gal80^ts^, an inhibitor of Gal4 function. In this way, we could express the *RNAi-opa^II^* at different developmental times before eclosion and score the percentage of adult flies presenting the phenotype described above. We noted that the earlier *opa* function is inhibited the higher is the penetrance of post-eclosion defects (figure 3*b'*). In other words, Opa does not seem to be required in a particular developmental step, but rather continuously. Similar results were obtained using *RNAi-opa^X^*, another *RNAi-opa* line (figure 3*c*).

In newly emerged wild-type adult flies the metathoracic and abdominal ganglions include 14 neurons expressing Burs [13,27]. *Burs*-expressing neurons correspond to CCAP motoneurons, which also express the Dachshund (Dac) marker [28]. Similar features are found in our control *CCAP-Gal4 UAS-GFP UAS-Dcr-2* flies (figure 3*d*). When we examined the VNC of *CCAP-Gal4 UAS GFP UAS-Dcr-2 RNAi-opa^II^* flies we noted an alteration in the driver expression due to the reduction of GFP signal. We also observed a decrease in the number of *Burs*-expressing neurons (figure 3*e*). Statistical analysis (figure 3*f*) revealed a threefold decrease of *Burs*-expressing neurons in this condition compared with control. This result

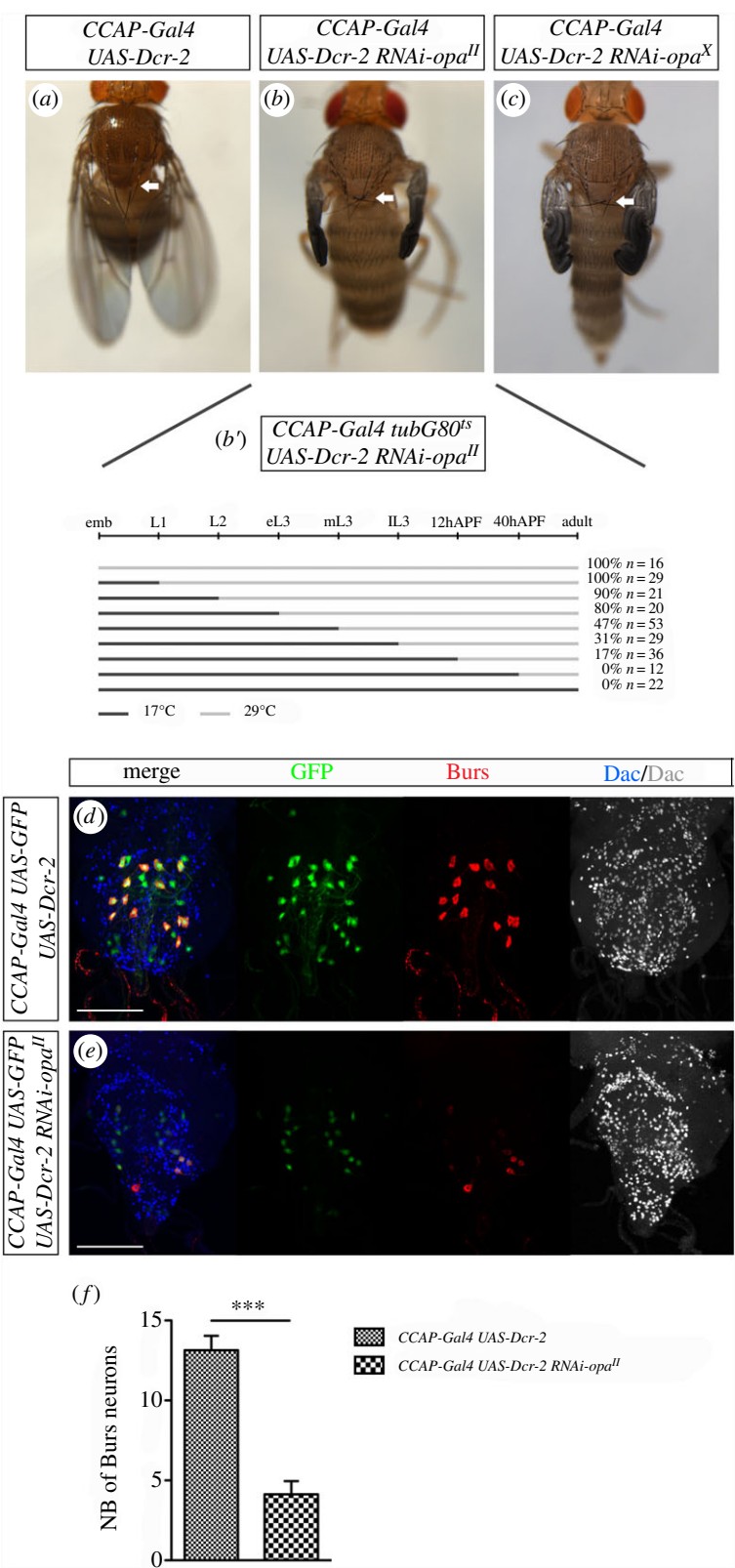

**Figure 3.** Knockdown of *opa* leads to defects in the post-ecdysis maturation. (*a*) Control condition of normal post-ecdysis sequence corresponding to the ectopic expression of *Dcr-2* in the CCAP neurons. The arrow point out the normal crossing of the post-scutellar bristles. (*b*,*b'*) Reduction of *opa* specifically in the CCAP neurons expressing *UAS-Dcr-2* and *UAS-RNAi-opa*[II] leads to adult flies that do not inflate wings, fail to stretch their thorax as evidenced by the pronounced crossing of the post-scutellar bristles (arrow) and do not show either a sclerotized or melanized cuticle (*b*) (100%, *n* = 150). The use of *tubG80*[ts] allows a temporally control the ectopic expression *RNAi-opa*[II] by a switch of temperature from 17°C ( permissive temperature) to 29°C (restrictive temperature). Change of temperature at different developmental times does not reveal a specific time requirement of *opa* in the CCAP neurons (*b'*). (*c*) Reduction of Opa specifically in the CCAP neurons by expressing another UAS-*RNAi-opa*, referred as *RNAi-opa*[X], together with *UAS-Dcr-2* (100%, *n* = 120) shows the same defects than those shown in (*b*). The pictures of adult flies in (*a–c*) have been taken 3 h after eclosion. (*d*) VNC expressing *Dcr-2* in the CCAP cells is used as control. The expression of *CCAP* using the *CCAP-Gal4* driver and immunostained for Burs and Dac are shown. (*e*) Reduction of *opa* expression in the CCAP neurons leads to a decrease in *CCAP-Gal4* and Burs expression levels. VNC shown in (*d*) and (*e*) have been dissected from adult flies that were 0–1 h old. Scale bar, 100 μm. (*f*) Statistical analysis of number of cells expressing Burs. In *CCAP-Gal4 UAS-Dcr-2* flies, *n* = 7; mean of number of neurons expressing Burs per VNC = 13,1 (± s.e.m. 0,3). In *CCAP-Gal4 UAS-Dcr-2 UAS-RNAi-opa*[II] flies, *n* = 8; mean of number of neurons expressing Burs per VNC = 4,1 (± s.e.m. 0,3). ****p-value < 0.0001.

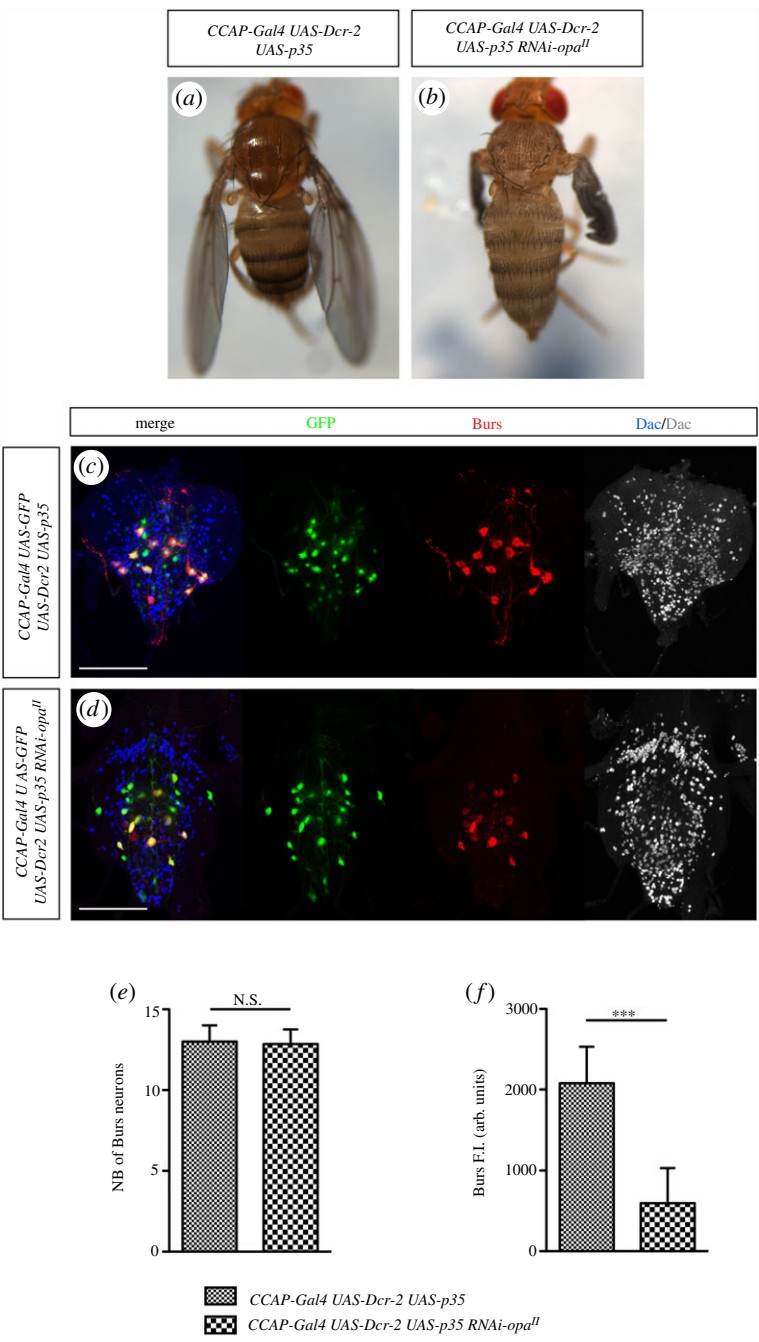

**Figure 4.** Opa regulates CCAP survival and Burs expression. (*a*) Ectopic expression of *p35* to block cell death together with Dcr-2 does not affect normal post-ecdysial maturation. (*b*) Simultaneous ectopic expression of *UAS-p35, UAS-Dcr-2 and UAS-RNAi-opa*[II] in the CCAP neurons does not rescue the post-eclosion phenotype induced by the lack of *opa* shown in figure 3*b* (100%, *n* = 60). The pictures of adult flies shown in (*a,b*) have been taken 3 h after eclosion. (*c*) VNC of control flies that express simultaneously *UAS-Dcr-2* and *UAS-p35* in the CCAP neurons. Inhibition of CCAP cell death has no effect on either CCAP cell number or *CCAP-Gal4* or Burs expression. (*d*) VNC of flies that express *UAS-p35, UAS-Dcr-2* and *UAS-RNAi-opa*[II] ectopically in the CCAP cells of the VNC show similar CCAP cell number than control flies (*c*). However, the expression of Burs significantly decreased compared with control. VNC in (*c,d*) have been dissected from adult flies that were 0–1 h old. Scale bar, 100 µm. (*e,f*) Statistical analysis of cells expressing Burs in *CCAP-Gal4, UAS-Dcr-2* and *UAS-p35* flies (*e*), *n* = 7; mean of number of neurons expressing Burs per fly VNC = 13,0 (s.e.m. ± 0,4) and for *CCAP-Gal4 UAS-Dcr-2 UAS-p35* and *UAS-RNAi-opa*[II] flies (*e*), *n* = 7; mean of number of neurons expressing Burs in each VNC = 12,9 (s.e.m. ± 0,3). No significant difference (N. S.) mean *p*-values ≥= 0.05. (*f*) Statistical analysis of cells expressing Burs in *CCAP-Gal4, UAS-Dcr-2, UAS-p35* flies, *n* = 88 cells; mean of Burs fluorescent intensity (F.I.) is in arbitrary units (a.u.) per cell = 2079 (s.e.m. ± 48). In *CCAP-Gal4 UAS-Dcr-2, UAS-p35, UAS-RNAi-opa*[II] flies, *n* = 85 cells; mean of Burs F.I. in a.u. = 596 (s.e.m. ± 47) per cell. ****$p$-value < 0.0001.

indicates an alteration in the execution of the post-ecdysis sequence in adult flies when *opa* expression is reduced in the CCAP cells (figure 3*b,c*).

To explain how Opa affects the expression of *CCAP* and Burs neuropeptides, we considered two not exclusive hypotheses. On the one hand, Opa could positively regulate the expression of CCAP and Burs neuropeptides; on the other hand, the decrease in number of cells expressing these

neuropeptides could be consequence of death of CCAP cells. To discriminate between these two possibilities, we used the cell death inhibitor *p35*. In our *CCAP-Gal4 UAS-Dcr-2 UAS-p35* control flies the post-eclosion maturation occurs normally (figure 4*a*), without consequences either in the number of CCAP cells or in the expression of *CCAP-Gal4* or Burs in recently born adults (figure 4*c*) [29]. When we simultaneously blocked apoptosis and decreased *opa* expression level in flies

royalsocietypublishing.org/journal/rsob    Open Biol. **9**: 190245

*CCAP-Gal4 UAS-Dcr-2 UAS-p35 RNAi-opa*[II] we were not able to see rescue of the adult mutant phenotype (figure 4b). However, the quantified number of CCAP cells is similar to control (figure 4c–e); this indicates that CCAP neurons enter in apoptosis when *opa* function is removed. Nevertheless, we noted that Burs expression in these brains (figure 4d) is lower than in the control situation (figure 4c). Quantification analysis revealed a threefold difference (figure 4f). Therefore, the lack of phenotypic rescue when *opa* function is removed while simultaneously cell death is blocked in CCAP cells (figure 4b) has to be due to a reduction of Burs expression. Similar results have been obtained when cell death was inhibited by knocking down the pro-apoptotic *hid*, *grim* and *rpr* genes with *UAS-RHG miRNA* expression (not shown). Altogether, these results indicate that Opa is needed not only to maintain CCAP viability but also to regulate *Burs* expression.

## 2.3. Ectopic expression of Opa or its homologue Zic2 is sufficient to activate Burs expression and disturbs adult post-ecdysis

We next wondered whether an excess of Opa expression in the CCAP cells could also disturb the ecdysis sequence. We observed that *CCAP-Gal4 UAS-opa* flies reached adulthood, but they presented characteristic alterations in the post-ecdysis sequence (figure 5b and a for comparison). All flies exhibit folded wings, a lack of thorax extension and defective cuticle sclerotization, as shown by their dull aspect, although melanization seems to be normal. Similar post-ecdysis defects were obtained when ectopically expressing Zic2 in the CCAP neurons (figure 5c), although the penetrance of the phenotype is incomplete: 15% show a strong phenotype, another 15% a mild phenotype and the remaining 70% show a normal morphology (not shown).

In control adult VNC, expression of Burs is restricted to the CCAP motoneurons T3-A7 (Dac+) and is absent from the CCAP interneurons and the A8-A10 (Dac-) [30]. Thus, Burs is expressed in 14 CCAP out of 32 CCAP cells in neuromeres from T3 to A10 (figure 5d). Although the total number of CCAP cells in adult brains of flies overexpressing Opa in the entire CCAP population is similar to controls, there is a change in the normal expression of the neuropeptides. The *CCAP* reporter expression decreases in all cells and Burs is now detected in the more posterior motoneurons and interneurons, which do not normally express Burs (figure 5e,g for the statistical analysis). The CCAP interneurons seem to maintain the interneuron fate given that they do not express Dac (compare inserts in figure 5d with e). Ectopic Zic2 in the CCAP cells (with a strong post-eclosion phenotype) present the same characteristics (figure 5f,g for the statistical analysis). Altogether, these results indicate that overexpression of Opa or ectopic expression of Zic2 negatively represses the *CCAP* neuropeptide expression and, conversely, positively regulates Burs expression. Given that the neuropeptide CCAP is not implicated in adult ecdysis [31], we propose that the post-ecdysis phenotype, in this case, is due to an excess of Burs in the CCAP interneurons.

Next, we wanted to test if the capability of Opa to activate Burs expression is specific to the CCAP neurons or not. To this aim, we first expressed Opa in all neuronal cells using *elav-Gal4* driver. Since these individuals die during embryogenesis, we used *tubG80*[ts] to initiate ectopic expression during larval

stages. When Opa is expressed ectopically for 4 days, flies do not reach adulthood and die during metamorphosis or even at earlier stages (figure 6a). Larval CNS shows ectopic expression of Burs, especially in ventral and lateral neurons of the thoracic and more anterior abdominal segments (figure 6b,c for control expression). Because expressing ectopically Opa during a longer time is lethal, we could not test whether Burs could be ectopically expressed in more neurons by increasing Opa expression time. We then expressed Opa under the control of a peptidergic line *36Y-Gal4* and also overcoming the embryonic lethality associated with Opa gain-of-function expression using *tubG80*[ts]. After 3 days of Opa ectopic expression during the larval period, animals die in the early stages of metamorphosis or during the late larval stage (figure 6d). Interestingly, these larvae have double sets of mouth hooks (figure 6e, arrow; figure 6f for wild-type), a typical moulting defect. Opa was also able to ectopically activate Burs in several, but not all, *36Y-Gal4*-expressing neurons in the lateral VNC sides (figure 6g). Taking all these results together, we conclude that Opa is able to activate Burs expression in VNC neurons other than CCAP.

## 2.4. Peripheral CCAP neurons also express *opa*

In *Drosophila*, CCAP neurons are not restricted to the CNS. Indeed, we observed several somas of CCAP neurons on the ventral part of the adult abdomen. These neurons are tightly associated with the pleural internal transverse muscles (PITM) [32] (figure 7A,A′). These muscles are present at the border of each abdominal segment and have elongated fibres perpendicular to the midline, which extend from the ventral midline to the lateral sides. For each PITM muscle observed, either one CCAP body cell or none is detected nearby, although we did not find a specific pattern of these peripheral CCAP neurons. These peripheral CCAP neurons seem to be in contact with the CCAP axons from the VNC (figure 7a). Peripheral CCAP neurons innervating the dorsal vessel present on the lateral sides of the adult abdomen (figure 7b) have been already reported [33,34]. We have also identified new CCAP cell populations located on the posterior part of the adult abdomen and associated with the posterior digestive tract and the sexual organs. Several CCAP cells are connected with the muscles of the rectal ampulla (figure 7c,g). In females, a peripheral CCAP surrounds the oviduct (figure 7d,h) and the spermathecae (figure 7e). In males, the inner face of the ejaculatory bulb is highly innervated by a peripheral CCAP neuron (figure 7f). Finally, we analysed if these peripheral CCAP neurons expressed *opa* and we have observed that all of them did (figure 7A–f), consistent with the observed *opa* expression in all the CCAP neurons of the CNS. Interestingly, Burs can be detected along the axons of these CCAP peripherical neurons (figure 7g–h′). Further studies are required to understand the role of the neuropeptide Burs in these cells.

## 3. Discussion

### 3.1. Opa as a marker of the CCAP neurons in the CNS and PNS

So far, few markers have been identified to be expressed in the CCAP neurons—Dac, nuclear phosphorylated Mad (pMad) and *Ok6-Gal4*—but their expression is limited to the CCAP

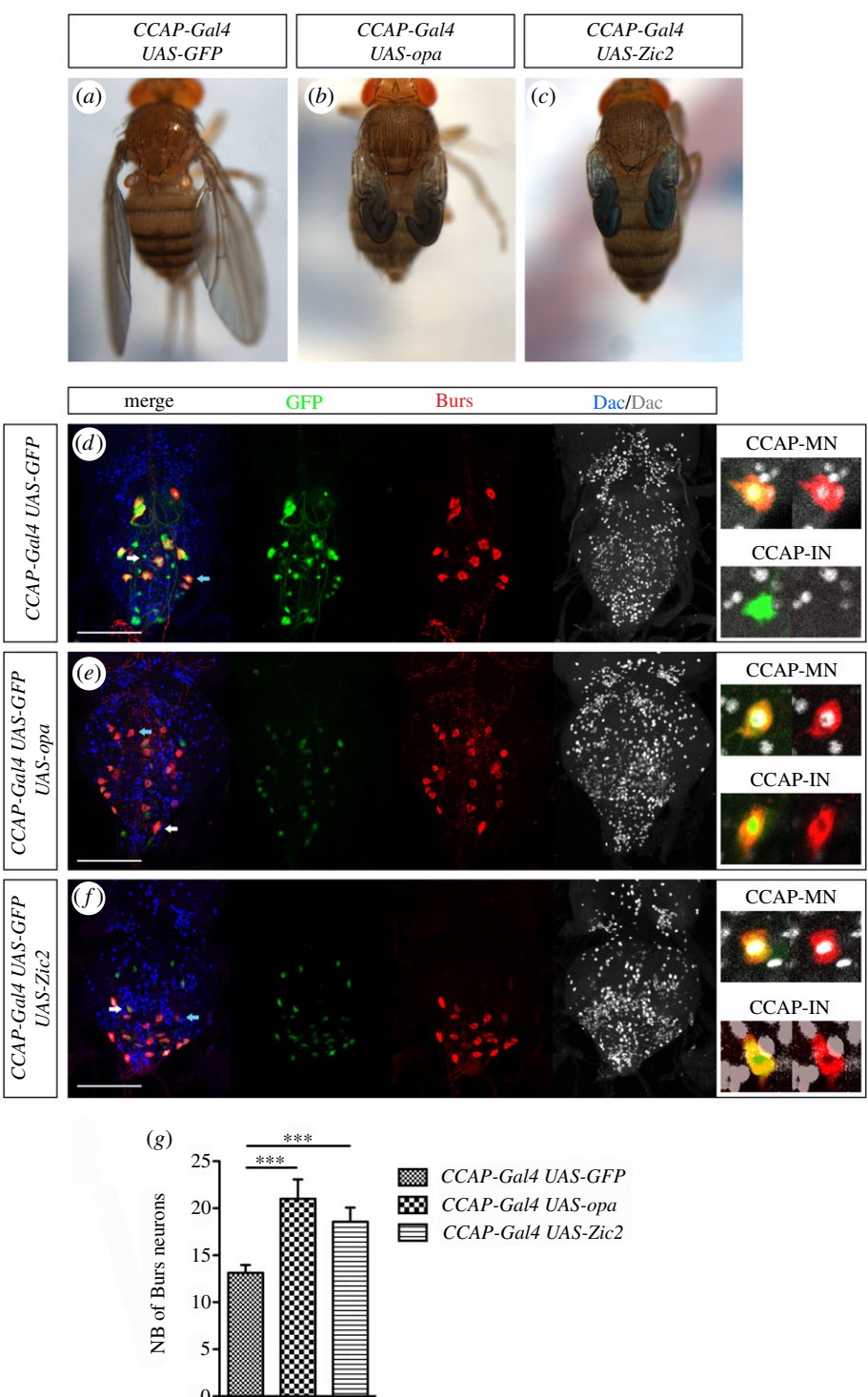

**Figure 5.** Overexpressing Opa or ectopic expression of Zic2 in the CCAP neurons ectopically activates Burs expression in the CCAP interneurons and affects post-ecdysis maturation. (*a*) Adult *CCAP-Gal4; UAS-GFP* flies are used as controls. (*b*) Overexpression of Opa in the CCAP neurons causes defects in adult ecdysis: wings do not expand, the thorax fails to stretch, the cuticle is matte but the pigmentation is similar to control (100%, *n* = 83). (*c*) Ectopic expression of Zic2 is also able to alter the post-ecdysis maturation but with a less frequency (30%, *n* = 93). Pictures in (*a–c*) were taken from 3 h-old adults. (*d*) Control adult VNC expressing *CCAP-Gal4 UAS-GFP* and stained for Burs and Dac. (*e*) Overexpression of Opa in the CCAP neurons (same genotype as in (*b*)) leads to a general decrease of *CCAP* expression, but also to an ectopic expression of Burs in the CCAP interneurons. (*f*) Flies with the stronger phenotype after ectopic expression of Zic2 in the CCAP neurons (VNC are dissected from flies of the genotype presented in (*c*)). Flies have similar alterations as the ones shown in (*e*). VNC in (*e,f*) were dissected from 0 to 1 h-old adults. The inserts in (*d–f*) are zoom of a CCAP motoneuron (CCAP-MN, marked by a blue arrow) and a CCAP interneuron (CCAP-IN, marked by a white arrow in the merge panels). Scale bar, 100 μm. (*g*) Statistical analysis of cells expressing Burs in *CCAP-Gal4 UAS-GFP* flies, *n* = 8; mean of number of neurons expressing Burs per VNC = 13,1 (s.e.m. ± 0,3). And in *CCAP-Gal4 UAS-opa* flies, *n* = 7; mean of number of neurons expressing Burs per VNC = 21,0 (s.e.m. ± 0,8). For *CCAP-Gal4 UAS-Zic2*, *n* = 7; mean of number of neurons expressing *Burs* per VNC = 18,6 (s.e.m. ± 0,6). ****$p$-value < 0.0001.

motoneurons and they have the disadvantage to be expressed in many more neurons throughout the VNC [28]. In this work, we describe that *opa-lacZ* is expressed in all the CCAP cells of the CNS, motoneurons and interneurons, from the embryonic stage 12 to the adult (electronic supplementary material, figure S1; figure 1), indicating that *opa* is expressed throughout

royalsocietypublishing.org/journal/rsob    Open Biol. 9: 190245

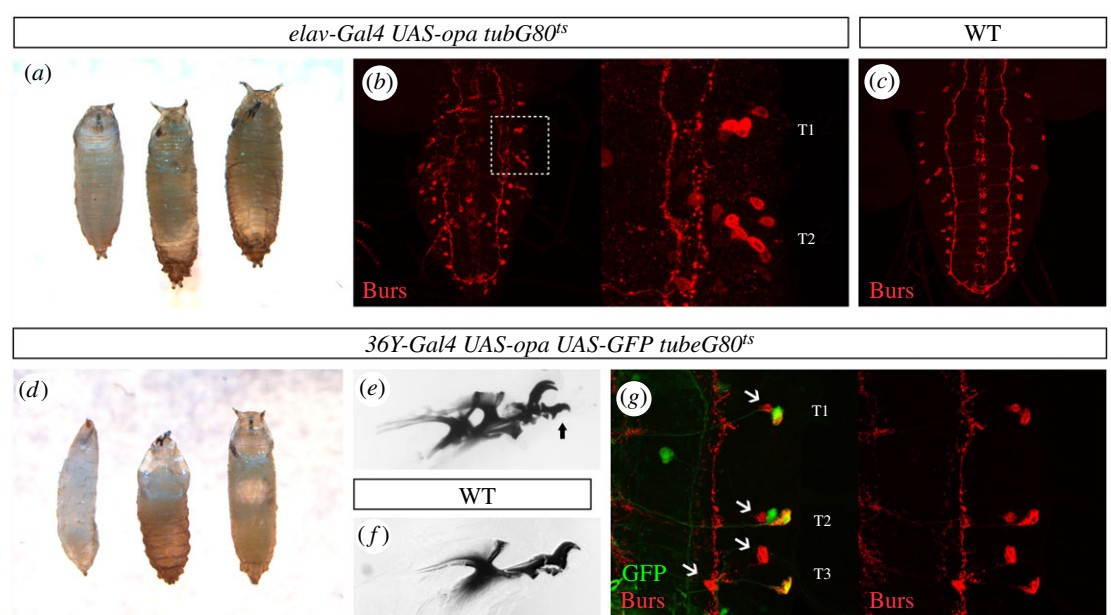

**Figure 6.** Opa activates Burs expression in no-CCAP neurons. (*a*) Ectopic expression of *opa* under the control of *elav-Gal4* leads to early pupal lethality. (*b*) Larval CNS of the previous genotype showing an ectopic activation of Burs in a large number of neurons, especially in lateral neurons of the thoracic VNC. (*c*) Wild-type expression of Burs in the VNC of an L3 larva. At this stage, hemineuromeres T1 and T2 express Burs in one neuron and hemineuromeres T3-A7 express Burs in two cells. (*d*) Ectopic expression of *opa* under the control of *36Y-Gal4* also leads to early pupal lethality. (*e*) Larvae of a similar genotype showing that the L2 mouth parts (arrow) do not detach from the L3 mouth hooks. (*f*) Wild-type mouth hooks. (*g*) Ectopic *Burs* activation can be observed in a subset of *36Y-Gal4 UAS-opa* expressing cells in the lateral VNC. The arrowheads indicate the endogenous expression of Burs. (*b,c,g*) correspond to the maximum projection of stacks.

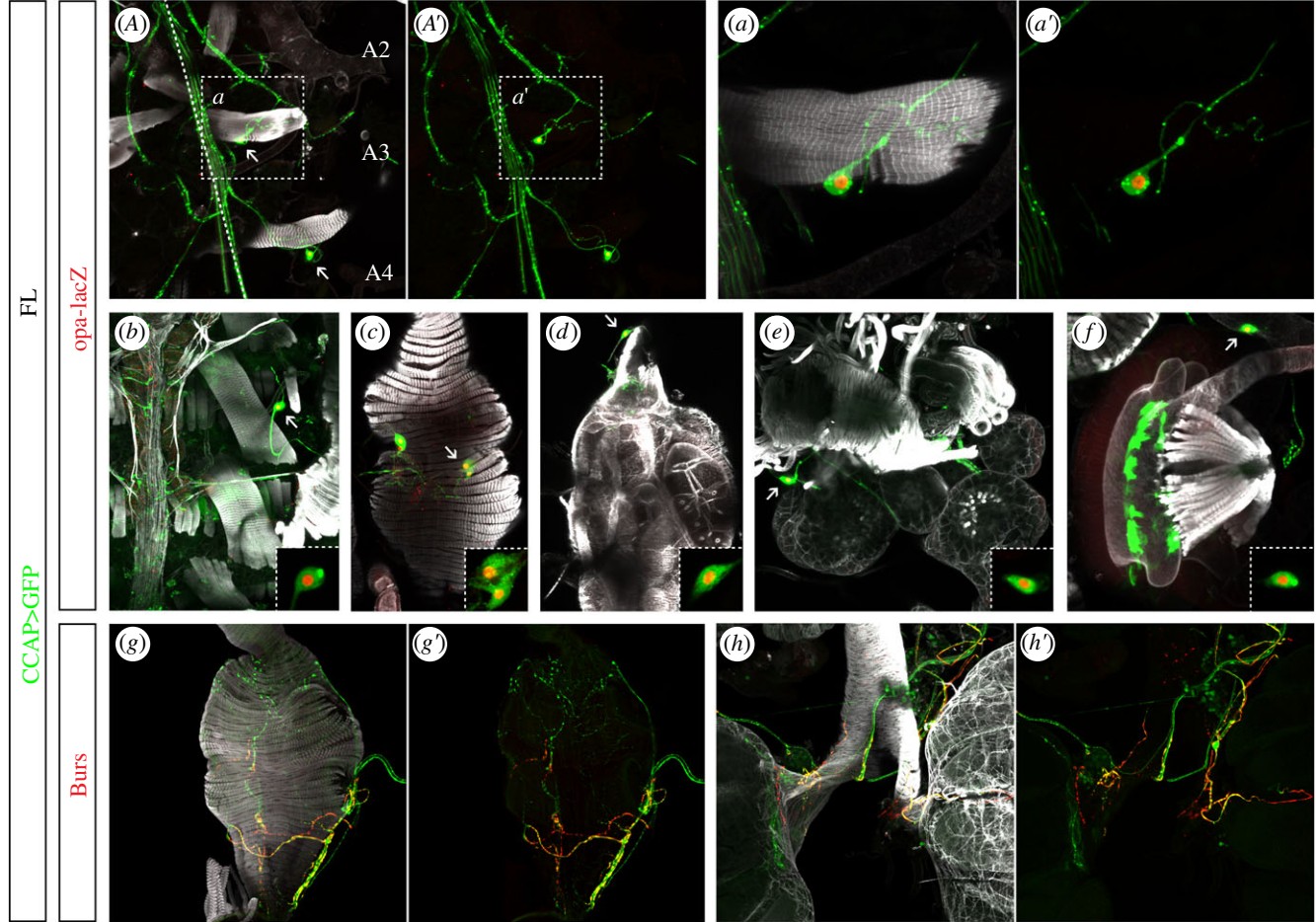

**Figure 7.** Expression of *opa* and Burs in adult peripheral CCAP neurons. (*A–a′*) CCAP neurons innervating the pleural internal transverse muscles (PITM); the adult muscles are inherited from the larva. The picture is taken at the level of the PITM at the A2/A3 and A3/A4 borders. Both PITM are associated with a CCAP cell body (arrows). (*a*) and (*a′*) are zoom from the marked square region in (*A*) and (*A*), respectively. The dashed line marks the ventral midline with the anterior towards the top. (*b–h′*) CCAP neurons innervating the aorta (*b*), the rectal ampulla (*c,g,g′*), the oviduct (*d,h,h′*), the spermatheca (*e*) and the ejaculatory bulb (*f*). All peripheral CCAP neurons expressed *opa* (*b–f*) and Burs (*g–h′*). The inserts in (*b–f*) correspond to CCAP cell bodies marked by arrows. Pictures have been taken from recently born adults. Phalloidin (FL) marks filamentous actin. All images are maximum projections of stacks except the inserts that correspond to a single section.

royalsocietypublishing.org/journal/rsob    Open Biol. 9: 190245

the lifetime of the CCAP neurons. In the dorsal part of the VNC, *opa-lacZ* expression is exclusive to the CCAP neurons (figure 2*a*); the rest of the *opa*-expressing cells are found mainly in the ventralmost part of the VNC. We identified some of these cells as peptidergic cells (figure 2*b*). Besides, *opa-lacZ* is also expressed in peripheral CCAP neurons (innervating the aorta, the digestive system, the sexual organs and the doomed adult muscles) (figure 7). Therefore, Opa seems to be an appropriate marker for the CCAP cells.

## 3.2. Cell death in the NB 3–5 lineage

*Drosophila* CNS is constantly subjected to neuronal death. Usually, a given NB generates the same lineage in all the neuromeres, and cell death is in charge of discarding the unnecessary neurons on each specific neuromere. This phenomenon occurs with the lineage of the NB 3–5 that gives rise to one CCAP motoneuron and one CCAP interneuron in each hemineuromere from SE1-A7. In the case of SE1-T2 neuromeres, CCAP motoneurons undergo programmed cell death as soon as they differentiate [11] and it has been shown that ectopic expression of the Hox gene *Ubx* is sufficient to prevent apoptosis of these cells [35]. It has also been described that Ubx prevents death of CCAP motoneurons in the T3-A1 segments and Abd-A in the A2-A7 segments [35]. Later on, CCAP neurons progressively degenerate in adults as soon as they have accomplished their function [13].

We have observed here that *opa* downregulation in the CCAP neurons causes a significant decrease in their number (figure 3*e*). Since this number is recovered when apoptosis is blocked (figure 4*d*), we investigated if Opa could act as a CCAP survival factor. Interestingly, we observed that controlling the timing of *opa* downregulation does not reveal an Opa requirement at any particular developmental stage, meaning that CCAP neurons constantly need Opa expression to survive.

In the zebrafish *Dario rerio*, it has been shown that depletion of the Opa homologue Zic1 induces cell death in the prospective diencephalon [36] and knockdown of Zic2 leads to cell death during neuronal differentiation [37], indicating a conservation of Zic function in preventing neural cell death in vertebrates.

## 3.3. Opa is an activator of Burs expression

In addition to the function of Opa in CCAP cell viability, we found that Opa acts as a positive regulator of Burs expression. In thoracic and abdominal segments of wild-type adults, Burs expression is restricted to the CCAP motoneurons T3-A7 [30]. When *opa* function is reduced in CCAP cells and cell death prevented, a decrease in the number of *Burs*-expressing cells is still observed in the newly formed adult CNS (figure 4*d*). Conversely, when Opa is overexpressed in the CCAP cells, Burs is detected in more posterior motoneurons and interneurons (figure 5*e*). Although weaker, similar effects have been obtained by expressing Zic2 ectopically in the CCAP neurons (figure 5*f*), revealing an underlying conserved mechanism of action of Zic proteins.

Opa is also able to activate Burs expression in neurons other than CCAP because excess of Opa expression under the control of either a pan-neuronal or a peptidergic line also activates Burs in no-CCAP neurons (figure 6*b,g*). However, our experiments indicate that Opa is not sufficient to activate Burs expression in all neurons. Our interpretation is

that the competence to activate Burs expression could have been lost in some aged neurons or alternatively, it could depend on the neuronal type, some being more sensitive to respond to Opa ectopic expression than others.

*Burs* is expressed in the CCAP interneurons during larval stages and its expression is turned off during metamorphosis [27,29,30]. Although the reason for this change has not yet been determined, according to our results Opa could activate *Burs* expression in the interneurons during early stages and, at a later stage, a hypothetic repressor could prevent the expression of *Burs*. Alternatively, the lack of *Burs* expression in the adult CCAP interneurons could be due to the lack of an Opa coactivator still unknown.

## 3.4. Anatomical constrains can explain the presence of peripheral CCAP neurons in *Drosophila melanogaster*

During metamorphosis, most of the larval muscles are hydrolysed and adult muscles are formed de novo. The larval PITM, however, escape this degradation and are maintained until the first 12 h of adulthood, when the fly abdomen has its definitive muscular pattern [32]. Because of the timing of degradation, these larva-inherited adult muscles are believed to be required for adult ecdysis [32]. Interestingly, we often found that peripheral CCAP neurons innervate these doomed muscles (figure 7*a,a'*), in addition to the CCAP neurons of the CNS that also might directly innervate them. Besides, it has been described that abdominal contractions generate a haemolymph flux allowing wing extension and thorax expansion after eclosion [5,38]. The Opa ecdysis phenotype suggests that the peripheral CCAP neurons might mediate the PITM contractions to generate internal pressure of haemolymph flux during eclosion. This opens up the possibility that the post-ecdysis maturation of wing extension and thorax stretching could be a requirement of Opa only in peripheral CCAP neurons.

To our knowledge, no peripheral CCAP neurons have been reported in other species. In *Drosophila*, the release of the *CCAP* neuropeptide by peripheral CCAP neurons is known to mediate dorsal vessel contraction [33,39,40]. This role of *CCAP* neuropeptide as heart contraction modulator is conserved in other arthropod species [41,42]. In the case of *Manduca sexta*, it has been described that the CCAP neurons from the CNS directly innervate the dorsal vessel [43].

Unlike other model organisms used to study arthropod ecdysis, we found that in *Drosophila* the peripheral CCAP cells innervate the posterior digestive tract and the sexual organs (figure 7*b–h*). In the locust *Locusta migratoria*, CCAP neuropeptide stimulates contractions of both the spermatheca and the gut by direct innervations of CCAP neurons of the CNS [44,45]. In *M. sexta*, CCAP also stimulates contractions of the oviducts, acting as neurohormone since the oviduct is not innervated by CCAP neurons [46]. In flies, CCAP may act as a neuromodulator being delivered directly to the oviduct by the peripheral CCAP neuron.

Anatomical differences could explain the lack of peripheral CCAP neurons in some species. Thus, in *L. migratoria* or *M. sexta*, the VNC expands all along the body permitting the proximity between the CCAP neurons of the VNC and abdominal target organs. In *Drosophila*, the posterior abdomen is located far enough from the VNC to explain the

need to have peripheral CCAP neurons in close contact to the posterior digestive tract and sexual organs.

# 4. Conclusion and perspective

Our work reveals that the transcription factor Opa in *Drosophila* is an ecdysis regulator, intervening in different step of this process. First, loss and gain of function of *opa* are able to missregulate Burs expression, a conserved neuropeptide involved in ecdysis in arthropods (reviewed in [47,48]). Second, Opa prevents CCAP neurons from cell death, both in the embryo and in post-embryonic stages. Finally, *opa* is expressed in peripheral CCAP neurons innerving the doom muscles involved in adult ecdysis.

A recent screening by dsRNA injection looking for genes implicated in the metamorphosis of the red flour beetle *Tribolium castaneum* identified Opa as the only transcription factor candidate [49], indicating a possible conserved function of Opa in insects. It would be very interesting to analyse whether *opa* is expressed not only in other insects but also in other arthropods, and if *opa* activities to regulate apoptosis and *Burs* expression are also conserved.

# 5. Material and methods

## 5.1. Fly stock

*RNAi-opa* (from VDRC #101531, here called *RNAi-opa^II^*, and VDRC #51292, here called *RNAi-opa^X^*); *opa^7^* [15]; *opa-lacZ* (called 3–66 in [19]; the location of the p-element insertion is described in [50], *odd-lacZ* [51], *CCAP-Gal4* [5], *386Y-Gal4* [24], *36Y-Gal4* (Min), *UAS-RHG miRNA* [52], *hs-flp; tub > GFP > Gal4* [53], *elav-Gal4, UAS-p35, UAS-Dcr-2, UAS-GFP, tub-Gal80^ts^, hs-flp; FRT82B Ubi-GFP* (BDSC).

## 5.2. Immunohistochemistry

First antibodies used: rabbit anti-β-gal 1 : 1000 (Jackson Laboratories), mouse anti-Dac 1 : 50, (Iowa Hybridoma bank), rabbit anti-Burs 1 : 2000 (Peabody *et al*. [27]). Secondary antibodies used: goat anti-rabbit conjugated with TRITC and goat anti-mouse conjugated with Cy5 1 : 200 (Jackson Laboratories). Phalloidin (FL)-TRITC 1 : 2000 (Sigma) was used to mark actin filaments and it was added together with the secondary antibodies.

## 5.3. Construction of *UAS-opa and UAS-Zic2*

To construct the UAS-Opa, the *opa* cDNA was amplified by PCR from LD30441 clone (Berkeley Drosophila Genome Project Gold cDNA; Drosophila Genomics Resource Center) using the sense primer: 5′ CACCATGATGATGAACGCCTTCATT 3′ and the antisense primer: 5′ ATACGCCGTCGCTGCGCC 3′, cloned in the entry vector *pENTR/D-TOPO* (Gateway system; Invitrogen) and introduced by recombination in the destination vector *pTWH*.

To construct the UAS-Zic-2, the *zic-2* cDNA from *Xenopus laevis* [54] (a gift from Ariel Ruiz i Altaba) was cloned using the polylinker restriction sites of the pUAST vector.

## 5.4. Images acquisition and processing

Images of fluorescent labelled tissues were taken with a Zeiss Laser Scanning Confocal Spectral LSM710 with Axiolmager M2 upright microscope. Images of adult wings and mouth hooks of larvae were taken with a Zeiss Axioskop upright microscope coupled with a monochrome CCD camera. Images of adult flies and pupae were taken with a Leica M205 FA microscope coupled with a DFC 550 camera. Images were processed with Adobe Photoshop CS7 and ImageJ 1.46.

## 5.5. Statistical analysis

Statistical analysis was performed with Student's *t*-tests using GraphPad Prism 5.0a. For all analyses, the s.e.m. (standard error of the mean) is shown.

Data accessibility. This article has no additional data.

Authors' contributions. E.S. and I.G. designed experiments, interpreted the data and wrote the manuscript. E.S. and S.F.d.l.P. performed experiments.

Competing interests. The authors declare no competing or financial interests.

Funding. This work was supported by the grant BFU (BFU2017-83789-P) from the Spanish Ministry of Economy and Competitiveness (MINECO), by a Marie Curie Action (FP7-2009 Ref: ITN 238186) to I.G. and by institutional grants from the Fundación Areces and from Banco de Santander to the CBMSO.

Acknowledgements. We thank Christian Klämbt for the *opa* enhancer trap line, Laura Torroja for advice and Pedro Ripoll for corrections on the manuscript. We are also very grateful to Ariel Ruiz i Altaba for the *Xenopus* Zic-2 cDNA and for his initial input to analyse the function of Opa and Zic2 during *Drosophila* late development. We also thank the anonymous reviewers for their comments, which have helped to improve this manuscript.

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
