## [Reviewer comments · Open Biology]

Review History

RSOB-18-0117.R0 (Original submission)

Review form: Reviewer 1

Recommendation

Major revision is needed (please make suggestions in comments)

Are each of the following suitable for general readers?

- a) **Title**
Yes
- b) **Summary**
Yes
- c) **Introduction**
Yes

Is the length of the paper justified?

Yes

Should the paper be seen by a specialist statistical reviewer?

No

Is it clear how to make all supporting data available?

Not Applicable

Is the supplementary material necessary; and if so is it adequate and clear?

Yes

Do you have any ethical concerns with this paper?

No

Comments to the Author

The manuscript by Simon et al. characterizes the expression and functional importance of the transcription factor *opa* in CCAP-expressing neurons of *Drosophila*. The manuscript reports several interesting observations that implicate *opa* in regulating the viability of CCAP-expressing neurons and their expression of the hormones CCAP and bursicon. In addition, the manuscript describes several previously unreported sites of expression of the CCAP-Gal4 driver in cells outside of the CNS. In general, the finding that *opa* is consistently expressed in CCAP neurons both inside and outside the CNS argues for a regulatory role for *opa* in the function of these cells, a conclusion also supported by the effects of *opa* downregulation via RNAi in CCAP neurons.

Although the presentation of the results is relatively straightforward, there are several issues that weaken the claims of the manuscript. The most important is that the only marker used to monitor *opa* expression is a lacZ enhancer-trap line that is very poorly characterized and has never been shown to faithfully replicate the expression pattern of the *opa* gene. The manuscript states that the "localization" of the lacZ line (see l. 259) is described in a PhD thesis, but I could find no evidence of this in the thesis after locating it online or in other published work. The expression pattern of the line outside of glia seems to be unpublished and the authors need to provide evidence that "*opa-lacZ*" actually reports the pattern of *opa* expression. Immunolabeling with anti-*opa* antibodies or comparison with the expression patterns of other lines (such as the *opa-lacZ* line published by Cimbora & Sakonju in 1995) are possible approaches. The observation that *opa*-RNAi expression in CCAP neurons generates mutant phenotypes is consistent with the expression of *opa* in at least some of these neurons, but does not strongly support the claim that *opa* is expressed in all of them. Other concerns are as follows.

Major Concerns:

1. The authors restrict their attention to CCAP and Burs neurons of the VNC, but why is not obvious. Confusingly, they also state (l. 180) that expression of Burs is restricted to T3-A10 in the VNC, but the paper they cite describes expression also in the subesophageal ganglion. Why are these neurons not mentioned, and why did the authors restrict their analysis to CCAP and Burs neurons in the VNC and not include the neurons of the SEZ and central brain. Also, in Fig.s 1 and 2A, the CCAP neurons of the central brain don't even seem to be labeled by *opa-lacZ*, which runs counter to the manuscripts conclusions. Are these neurons an exception, or is *opa-lacZ* labeling just very dim in them?
2. The experiments analyzing the effects of *opa* knockdown on the survival of CCAP and Burs neurons are somewhat confusing and inconclusive (see l. 147ff and Fig.s 3&4). This is because the only assessment of survival is GFP expression or Burs immunolabeling (i.e. there is no direct

assay for cell death). It could thus be that RNAi-*opa* reduces CCAP and Burs gene expression – sometimes below the signal detection threshold, but without killing the cells – and that UAS-*p35* completely restores the levels of CCAP, but not Burs expression. Evaluation of the results is also hampered by the fact that there is no explicit control demonstrating that UAS-*p35* is inhibiting apoptosis. Since the latter reagent is novel, a positive control demonstrating its efficacy is really necessary, although even such a demonstration would not rule out the possibility that it is merely regulating peptide expression (rather than apoptosis) in the case of the CCAP/Burs neurons.

4. Although the fidelity of expression of the CCAP-Gal4 line has been well established in the CNS by immunostaining, its fidelity outside of the CNS has not. While it is extremely likely that the novel CCAP-Gal4 cells described in the manuscript do in fact express CCAP, it would be desirable if this were either directly confirmed by immunostaining or the conclusions of the paper on this point tempered.

Minor comments (with reference to particular lines in the text or to the figures):

l. 93-4 The evidence that the late differentiating CCAP neurons are necessary and sufficient for pupal ecdysis, as originally reported, is inconsistent with results reported more recently by Diao et al. (*Genetics*, 2016). It would be useful to mention both sets of results.

l. 125 “miss position” What is meant here is unclear. Also, an arrow in the figure panel pointing to the bristle phenotype would be useful.

l. 136 “are composed of” should read “include”

l. 143 The decrease (13-→4) looks to be 3-fold not 4-fold

l. 144-5 Here and elsewhere the authors make an assertion about behavior (i.e. “alteration in the execution of the postecdysis sequence”) when all they appear to have examined is the endpoint (expansion of the wings). Since expansion failure can have causes other than behavioral deficits, and successful expansion can occur even in the absence of normal behavior. Some caution should therefore be exercised in the wording to indicate only that the end phenotype is either different or normal.

l. 159 “the quantified number of CCAP cells” presumably Burs cells are meant since this is what is reported in Fig. 4, but it would be interesting to know if here and in Fig. 3 the numbers of neurons that express CCAP, but NOT Burs is also reduced by RNAi-*opa* expression.

l. 178 What is meant by a “mild” phenotype?

l. 212 The ability of *opa* to activate Burs expression is actually rather limited based on the data presented and the conclusion that it does so “in many but not all VNC cells” seems exaggerated.

l. 223 and Fig. 7E: the text indicates that only one neuron is labeled, but the amount of labeling in the figure makes it look like more. Is it really only a single neuron?

Fig. 2B mentions data from a line called “386-Gal4” and several neurons are said to express myomodulin based on a paper not cited in the references. The presumed paper, however, does not refer to the stated Gal4 line and the manuscript/figure provides no independent evidence that myomodulin is expressed in the indicated neurons. Also, the 386-Gal4 line is not listed in the Materials and Methods though a 386Y-Gal4 line is – citing however a different paper. From that paper, the connection between 386Y-Gal4 and myomodulin is also not apparent.

Fig. 3B': what criteria were used to select the eL3, mL3, and IL3 animals?

Review form: Reviewer 2

Recommendation

Major revision is needed (please make suggestions in comments)

Are each of the following suitable for general readers?

- a) **Title**
No
- b) **Summary**
Yes
- c) **Introduction**
Yes

Is the length of the paper justified?

Yes

Should the paper be seen by a specialist statistical reviewer?

No

Is it clear how to make all supporting data available?

Yes

Is the supplementary material necessary; and if so is it adequate and clear?

Not Applicable

Do you have any ethical concerns with this paper?

No

Comments to the Author

This work reports on the role of the gene *opa* paired (*opa*) during the postembryonic development of *Drosophila*.

The findings include:

- *opa* is expressed throughout post-embryonic development in central neurons that express the neuropeptide CCAP; it is also expressed in peripheral CCAP neurons of the adult fly.
- Knocking down *opa* in CCAP neurons causes post-emergence defects, including failures to correctly expand the wings and to pigment the cuticle (exoskeleton). These defects do not appear to be caused by a requirement for *opa* during a specific postembryonic period.
- Knocking down *opa* in CCAP neurons also reduces the number of CCAP neurons and the levels of bursicon, a neurohormone expressed in a subset of CCAP neurons. Co-expression of the anti-apoptotic gene, *p35*, rescues the number of CCAP neurons, but not the levels of bursicon expression.
- Overexpressing *opa* in CCAP neurons causes all CCAP neurons to express bursicon.
- Overexpressing *opa* in all neurons causes larval and pupal lethality and expression of bursicon in non-CCAP neurons.

This is an interesting report as it identifies opa as an important gene involved in regulating CCAP neuron fate/survival as well as bursicon expression. The work is well done and described. Its major problems lie in the documentation, as described in detail below.

Major issues

1- The images that show co-localization of CCAP>GFP + opa-LacZ (Figs 1, 2), CCAP>GFP + dac + bursicon-IR (Figs. 3, 4, 5) are too low resolution and are maximal projections. The authors need to include higher magnification, single plane, images of at least some of the neurons to document co-localization; this is especially true for adult preparations, where neurons occur in a “clump”. Also, the expression of dac (Figs. 3, 4, 5) is barely visible (and it isn’t really clear why dac expression is included). If the authors wish to include this gene, these images in particular need to be greatly improved.

2- The images that illustrate the post-emergence defects of the adults are not good enough quality. This is especially true of Figs. 3A-C, where a good part of the fly is out of focus. The corresponding images in Figs. 5A-C are better but not great. The age of the flies (number of hours after emergence) should be indicated in the legends for Figs 3A-C, 4A,B and 5A-C.

Minor issues

3- The title and the text state that opa controls “adult postecdysis behavior”, yet no evidence is presented that behavior itself is affected. The fact that wing inflation is defective could be due to defects in the eversion of the wing imaginal disc at pupation (see 4 below). In addition, cuticle pigmentation is not a behavior. So, unless the authors specifically looked at postecdysis behaviors (e.g., wing inflation behaviors) and report on them, I would re-word these statements saying that “opa plays a role in post-emergence maturation” or something to that effect.

4- The postemergence wing expansion defects reported could be caused by defects at pupation, as incorrectly everted wing discs at pupal ecdysis will also lead to defects in the adult wing expansion. Thus, the authors should examine whether the animals pupate correctly. This can easily be determined by looking at pharate adults. Defects at pupation cause wings and legs to be shorter than normal (and, in extreme cases, for the head to be incorrectly everted), due to defective head and imaginal disc eversion.

5- Results lines 127-135. Although the results do suggest that opa is required continuously for the expression of a normal post-emergence phenotypes, Figure 3 indicates that there might be a sensitive period between the 3rd instar and the early pupa, suggesting a possible role in pupal ecdysis (see point 4, above). It would be useful if the authors did the complementary experiment, of raising the temperature in a staggered series starting in the larva (and/or restricting it to the period around pupation).

6- CCAP neurons are marked by using reporter line. Thus, CCAP expression itself is not measured, but GFP. This needs to be corrected throughout the text, changing “CCAP expression” for “CCAP reporter expression”, or something to that effect.

7- Line 93-94: the findings of Veverlytsa and Allan (2012) have recently been put into question (Diao et al. (2016) Genetics, Vol. 202, 175–189)

8- The experiment shown in Figure 4 should really use the genotypes:

CCAP>opa RNAi + dcr2 vs CCAP>opa RNAi + dcr2 + p35,

in other words showing what happens when p35 is co-expressed with opa RNAi in CCAP neurons vs when opa RNAi is expressed alone (with dcr2) in CCAP neurons.

9- The stippling in Figs 3F, 4E, 4F and 5G is distracting and seems unnecessary; white, grey and black would be sufficient.

10- The English needs to be improved throughout the text. In particular:

+ title: “Implication ...” should be “Involvement ...”; although “Role of...” might be preferable/

+ line 62: not clear why “thus” is used. Sentences in line 62-65 are unclear.

+ lines 144-146 are a little unclear.

+ line 173, should say “folded wings”

- + legend to Fig. 3 should read "Knockdown of opa leads to ..." (vs "Lack of opa leads to ...").
- + legend to Fig. 5 should read "Overexpressing opa..." (vs "Overexpression opa...").

Decision letter (RSOB-18-0117.R0)

26-Jul-2018

Dear Dr Simon,

We are writing to inform you that the Editor has reached a decision on your manuscript RSOB-18-0117 entitled "Implication of the *Drosophila* Zic family member Odd-paired in adult postecdysis behavior", submitted to Open Biology.

As you will see from the reviewers' comments below, there are a number of criticisms that prevent us from accepting your manuscript at this stage. The reviewers suggest, however, that a revised version could be acceptable, if you are able to address their concerns. If you think that you can deal satisfactorily with the reviewer's suggestions, we would be pleased to consider a revised manuscript.

The revision will be re-reviewed, where possible, by the original referees. As such, please submit the revised version of your manuscript within six weeks. If you do not think you will be able to meet this date please let us know immediately.

When submitting your revised manuscript, please respond to the comments made by the referee(s) and upload a file "Response to Referees" in "Section 6 - File Upload". You can use this to document any changes you make to the original manuscript. In order to expedite the processing of the revised manuscript, please be as specific as possible in your response to the referee(s).

Please see our detailed instructions for revision requirements
<https://royalsociety.org/journals/authors/author-guidelines/>

Sincerely,

The Open Biology Team
mailto: openbiology@royalsociety.org

Reviewer(s)' Comments to Author(s):

Referee: 1

Comments to the Author(s)

The manuscript by Simon et al. characterizes the expression and functional importance of the transcription factor *opa* in CCAP-expressing neurons of *Drosophila*. The manuscript reports several interesting observations that implicate *opa* in regulating the viability of CCAP-expressing neurons and their expression of the hormones CCAP and bursicon. In addition, the manuscript describes several previously unreported sites of expression of the CCAP-Gal4 driver in cells outside of the CNS. In general, the finding that *opa* is consistently expressed in CCAP neurons both inside and outside the CNS argues for a regulatory role for *opa* in the function of these cells, a conclusion also supported by the effects of *opa* downregulation via RNAi in CCAP neurons.

Although the presentation of the results is relatively straightforward, there are several issues that weaken the claims of the manuscript. The most important is that the only marker used to monitor *opa* expression is a lacZ enhancer-trap line that is very poorly characterized and has never been shown to faithfully replicate the expression pattern of the *opa* gene. The manuscript states that the "localization" of the lacZ line (see l. 259) is described in a PhD thesis, but I could find no evidence of this in the thesis after locating it online or in other published work. The expression pattern of the line outside of glia seems to be unpublished and the authors need to provide evidence that "*opa-lacZ*" actually reports the pattern of *opa* expression. Immunolabeling with anti-*opa* antibodies or comparison with the expression patterns of other lines (such as the *opa-lacZ* line published by Cimbora & Sakonju in 1995) are possible approaches. The observation that *opa*-RNAi expression in CCAP neurons generates mutant phenotypes is consistent with the expression of *opa* in at least some of these neurons, but does not strongly support the claim that *opa* is expressed in all of them. Other concerns are as follows.

Major Concerns:

1. The authors restrict their attention to CCAP and Burs neurons of the VNC, but why is not obvious. Confusingly, they also state (l. 180) that expression of Burs is restricted to T3-A10 in the VNC, but the paper they cite describes expression also in the subesophageal ganglion. Why are these neurons not mentioned, and why did the authors restrict their analysis to CCAP and Burs neurons in the VNC and not include the neurons of the SEZ and central brain. Also, in Figs 1 and 2A, the CCAP neurons of the central brain don't even seem to be labeled by *opa-lacZ*, which runs counter to the manuscripts conclusions. Are these neurons an exception, or is *opa-lacZ* labeling just very dim in them?
2. The experiments analyzing the effects of *opa* knockdown on the survival of CCAP and Burs neurons are somewhat confusing and inconclusive (see l. 147ff and Fig.s 3&4). This is because the only assessment of survival is GFP expression or Burs immunolabeling (i.e. there is no direct assay for cell death). It could thus be that RNAi-*opa* reduces CCAP and Burs gene expression – sometimes below the signal detection threshold, but without killing the cells – and that UAS-p35 completely restores the levels of CCAP, but not Burs expression. Evaluation of the results is also hampered by the fact that there is no explicit control demonstrating that UAS-p35 is inhibiting apoptosis. Since the latter reagent is novel, a positive control demonstrating its efficacy is really necessary, although even such a demonstration would not rule out the possibility that it is merely regulating peptide expression (rather than apoptosis) in the case of the CCAP/Burs neurons.
4. Although the fidelity of expression of the CCAP-Gal4 line has been well established in the CNS by immunostaining, it's fidelity outside of the CNS has not. While it is extremely likely that the novel CCAP-Gal4 cells described in the manuscript do in fact express CCAP, it would desirable if

this were either directly confirmed by immunostaining or the conclusions of the paper on this point tempered.

Minor comments (with reference to particular lines in the text or to the figures):

l. 93-4 The evidence that the late differentiating CCAP neurons are necessary and sufficient for pupal ecdysis, as originally reported, is inconsistent with results reported more recently by Diao et al. (Genetics, 2016). It would be useful to mention both sets of results.

l. 125 “miss position” What is meant here is unclear. Also, an arrow in the figure panel pointing to the bristle phenotype would be useful.

l. 136 “are composed of” should read “include”

l. 143 The decrease (13->4) looks to be 3-fold not 4-fold

l. 144-5 Here and elsewhere the authors make an assertion about behavior (i.e. “alteration in the execution of the postecdysis sequence”) when all they appear to have examined is the endpoint (expansion of the wings). Since expansion failure can have causes other than behavioral deficits, and successful expansion can occur even in the absence of normal behavior. Some caution should therefore be exercised in the wording to indicate only that the end phenotype is either different or normal.

l. 159 “the quantified number of CCAP cells” presumably Burs cells are meant since this is what is reported in Fig. 4, but it would be interesting to know if here and in Fig. 3 the numbers of neurons that express CCAP, but NOT Burs is also reduced by RNAi-*opa* expression.

l. 178 What is meant by a “mild” phenotype?

l. 212 The ability of *opa* to activate Burs expression is actually rather limited based on the data presented and the conclusion that it does so “in many but not all VNC cells” seems exaggerated.

l. 223 and Fig. 7E: the text indicates that only one neuron is labeled, but the amount of labeling in the figure makes it look like more. Is it really only a single neuron?

Fig. 2B mentions data from a line called “386-Gal4” and several neurons are said to express myomodulin based on a paper not cited in the references. The presumed paper, however, does not refer to the stated Gal4 line and the manuscript/figure provides no independent evidence that myomodulin is expressed in the indicated neurons. Also, the 386-Gal4 line is not listed in the Materials and Methods though a 386Y-Gal4 line is – citing however a different paper. From that paper, the connection between 386Y-Gal4 and myomodulin is also not apparent.

Fig. 3B’: what criteria were used to select the eL3, mL3, and IL3 animals?

Referee: 2

Comments to the Author(s)

This work reports on the role of the gene *odd paired* (*opa*) during the postembryonic development of *Drosophila*.

The findings include:

- opa is expressed throughout post-embryonic development in central neurons that express the neuropeptide CCAP; it is also expressed in peripheral CCAP neurons of the adult fly.
- Knocking down opa in CCAP neurons causes post-emergence defects, including failures to correctly expand the wings and to pigment the cuticle (exoskeleton). These defects do not appear to be caused by a requirement for opa during a specific postembryonic period.
- Knocking down opa in CCAP neurons also reduces the number of CCAP neurons and the levels of bursicon, a neurohormone expressed in a subset of CCAP neurons. Co-expression of the anti-apoptotic gene, p35, rescues the number of CCAP neurons, but not the levels of bursicon expression.
- Overexpressing opa in CCAP neurons causes all CCAP neurons to express bursicon.
- Overexpressing opa in all neurons causes larval and pupal lethality and expression of bursicon in non-CCAP neurons.

This is an interesting report as it identifies opa as an important gene involved in regulating CCAP neuron fate/survival as well as bursicon expression. The work is well done and described. Its major problems lie in the documentation, as described in detail below.

Major issues

- 1- The images that show co-localization of CCAP>GFP + opa-LacZ (Figs 1, 2), CCAP>GFP + dac + bursicon-IR (Figs. 3, 4, 5) are too low resolution and are maximal projections. The authors need to include higher magnification, single plane, images of at least some of the neurons to document co-localization; this is especially true for adult preparations, where neurons occur in a “clump”. Also, the expression of dac (Figs. 3, 4, 5) is barely visible (and it isn’t really clear why dac expression is included). If the authors wish to include this gene, these images in particular need to be greatly improved.
- 2- The images that illustrate the post-emergence defects of the adults are not good enough quality. This is especially true of Figs. 3A-C, where a good part of the fly is out of focus. The corresponding images in Figs. 5A-C are better but not great. The age of the flies (number of hours after emergence) should be indicated in the legends for Figs 3A-C, 4A,B and 5A-C.

Minor issues

- 3- The title and the text state that opa controls “adult postecdysis behavior”, yet no evidence is presented that behavior itself is affected. The fact that wing inflation is defective could be due to defects in the eversion of the wing imaginal disc at pupation (see 4 below). In addition, cuticle pigmentation is not a behavior. So, unless the authors specifically looked at postecdysis behaviors (e.g., wing inflation behaviors) and report on them, I would re-word these statements saying that “opa plays a role in post-emergence maturation” or something to that effect.
- 4- The postemergence wing expansion defects reported could be caused by defects at pupation, as incorrectly everted wing discs at pupal ecdysis will also lead to defects in the adult wing expansion. Thus, the authors should examine whether the animals pupate correctly. This can easily be determined by looking at pharate adults. Defects at pupation cause wings and legs to be shorter than normal (and, in extreme cases, for the head to be incorrectly everted), due to defective head and imaginal disc eversion.
- 5- Results lines 127-135. Although the results do suggest that opa is required continuously for the expression of a normal post-emergence phenotypes, Figure 3 indicates that there might be a sensitive period between the 3rd instar and the early pupa, suggesting a possible role in pupal ecdysis (see point 4, above). It would be useful if the authors did the complementary experiment, of raising the temperature in a staggered series starting in the larva (and/or restricting it to the period around pupation).
- 6- CCAP neurons are marked by using reporter line. Thus, CCAP expression itself is not measured, but GFP. This needs to be corrected throughout the text, changing “CCAP expression” for “CCAP reporter expression”, or something to that effect.
- 7- Line 93-94: the findings of Veverlytsa and Allan (2012) have recently been put into question (Diao et al. (2016) Genetics, Vol. 202, 175-189)

8- The experiment shown in Figure 4 should really use the genotypes:

CCAP>opa RNAi + dcr2 vs CCAP>opa RNAi + dcr2 + p35,

in other words showing what happens when p35 is co-expressed with opa RNAi in CCAP neurons vs when opa RNAi is expressed alone (with dcr2) in CCAP neurons.

9- The stippling in Figs 3F, 4E, 4F and 5G is distracting and seems unnecessary; white, grey and black would be sufficient.

10- The English needs to be improved throughout the text. In particular:

+ title: "Implication ..." should be "Involvement ..."; although "Role of..." might be preferable/

+ line 62: not clear why "thus" is used. Sentences in line 62-65 are unclear.

+ lines 144-146 are a little unclear.

+ line 173, should say "folded wings"

+ legend to Fig. 3 should read "Knockdown of opa leads to ..." (vs "Lack of opa leads to ...").

+ legend to Fig. 5 should read "Overexpressing opa..." (vs "Overexpression opa...").

Author's Response to Decision Letter for (RSOB-18-0117.R0)

See Appendix A.

RSOB-19-0245.R0

Review form: Reviewer 1

Recommendation

Accept with minor revision (please list in comments)

Do you have any ethical concerns with this paper?

No

Comments to the Author

The authors have addressed most of my concerns. I would ask only that they make the following changes:

l76 and elsewhere: change "behavior" to "maturation" as suggested by Reviewer 2. (Failure of flies to secrete bursicon into the hemolymph can give rise to all the phenotypes observed without necessarily affecting postecdysis behavior, so it is best not to explicitly attribute the deficits to behavioral failure in the absence of further data.)

l355-6: Replace "its localization is described..." with "the location of the p-element insertion is described..."

l235-6: Say: "Further studies are required to confirm the presence of CCAP and to confirm the role of this neuropeptide in these cells."

Discussion, Section 1: Include a statement that you have been unable to directly confirm by either immunostaining or in situ labeling that the expression seen with opa-LacZ faithfully mimics the expression of the gene. (This can be coupled to a statement about why you think the labeling pattern is nevertheless faithful.)

Review form: Reviewer 2

Recommendation

Accept with minor revision (please list in comments)

Do you have any ethical concerns with this paper?

No

Comments to the Author

The authors have addressed most of my concerns. I would just insist that they not refer to the adults defects as being in "postecdysis behavior" since a) they did not examine the fly's behavior, only the terminal phenotype (unexpanded wings) b) there seems to be additional non behavioral post-eclosion defects including in cuticle hardening (and possibly also melanization; difficult to tell from the pictures).

Decision letter (RSOB-19-0245.R0)

30-Oct-2019

Dear Dr Simon,

We are pleased to inform you that your manuscript RSOB-19-0245 entitled "Drosophila Zic family member Odd-paired is needed for adult postecdysis behavior" has been accepted by the Editor for publication in Open Biology. The reviewer(s) have recommended publication, but also suggest some minor revisions to your manuscript. Therefore, we invite you to respond to the reviewer(s)' comments and revise your manuscript.

Please submit the revised version of your manuscript within 7 days. If you do not think you will be able to meet this date please let us know immediately and we can extend this deadline for you.

1) A text file of the manuscript (doc, txt, rtf or tex), including the references, tables (including captions) and figure captions. Please remove any tracked changes from the text before submission. PDF files are not an accepted format for the "Main Document".

2) A separate electronic file of each figure (tiff, EPS or print-quality PDF preferred). The format should be produced directly from original creation package, or original software format. Please note that PowerPoint files are not accepted.

3) Electronic supplementary material: this should be contained in a separate file from the main text and meet our ESM criteria (see <http://royalsocietypublishing.org/instructions-authors#question5>). All supplementary materials accompanying an accepted article will be treated as in their final form. They will be published alongside the paper on the journal website and posted on the online figshare repository. Files on figshare will be made available approximately one week before the accompanying article so that the supplementary material can be attributed a unique DOI.

Online supplementary material will also carry the title and description provided during submission, so please ensure these are accurate and informative. Note that the Royal Society will not edit or typeset supplementary material and it will be hosted as provided. Please ensure that the supplementary material includes the paper details (authors, title, journal name, article DOI). Your article DOI will be 10.1098/rsob.2016[last 4 digits of e.g. 10.1098/rsob.20160049].

4) A media summary: a short non-technical summary (up to 100 words) of the key findings/importance of your manuscript. Please try to write in simple English, avoid jargon, explain the importance of the topic, outline the main implications and describe why this topic is newsworthy.

Images

Data-Sharing

It is a condition of publication that data supporting your paper are made available. Data should be made available either in the electronic supplementary material or through an appropriate repository. Details of how to access data should be included in your paper. Please see <http://royalsocietypublishing.org/site/authors/policy.xhtml#question6> for more details.

Data accessibility section

Sincerely,

The Open Biology Team

<mailto:openbiology@royalsociety.org>

Reviewer(s)' Comments to Author:

Referee: 2

Comments to the Author(s)

The authors have addressed most of my concerns. I would just insist that they not refer to the adults defects as being in "postecdysis behavior" since a) they did not examine the fly's behavior, only the terminal phenotype (unexpanded wings) b) there seems to be additional non behavioral post-eclosion defects including in cuticle hardening (and possibly also melanization; difficult to tell from the pictures).

Referee: 1

Comments to the Author(s)

The authors have addressed most of my concerns. I would ask only that they make the following changes:

l76 and elsewhere: change "behavior" to "maturation" as suggested by Reviewer 2. (Failure of flies to secrete bursicon into the hemolymph can give rise to all the phenotypes observed without necessarily affecting postecdysis behavior, so it is best not to explicitly attribute the deficits to behavioral failure in the absence of further data.)

l355-6: Replace "its localization is described..." with "the location of the p-element insertion is described..."

l235-6: Say: "Further studies are required to confirm the presence of CCAP and to confirm the role of this neuropeptide in these cells."

Discussion, Section 1: Include a statement that you have been unable to directly confirm by either immunostaining or in situ labeling that the expression seen with opa-LacZ faithfully mimics the expression of the gene. (This can be coupled to a statement about why you think the labeling pattern is nevertheless faithful.)

Author's Response to Decision Letter for (RSOB-19-0245.R0)

See Appendix B.

Decision letter (RSOB-19-0245.R1)

12-Nov-2019

Dear Dr Simon,

We are pleased to inform you that your manuscript entitled "Drosophila Zic family member Odd-paired is needed for adult postecdysis maturation" has been accepted by the Editor for publication in Open Biology.

You can expect to receive a proof of your article from our Production office in due course, please

check your spam filter if you do not receive it within the next 10 working days. Please let us know if you are likely to be away from e-mail contact during this time.

Article processing charge

Please note that the article processing charge is immediately payable. A separate email will be sent out shortly to confirm the charge due. The preferred payment method is by credit card; however, other payment options are available.

Sincerely,

The Open Biology Team
mailto:openbiology@royalsociety.org

Appendix A

Dear Editor,

Please find below our responses to reviewers's comments that have helped us to improve the manuscript. We have included in our answers the changes introduced in the text and figures in the new version of the manuscript.

Referee: 1

1- The manuscript states that the "localization" of the lacZ line (see l. 259) is described in a PhD thesis, but I could find no evidence of this in the thesis after locating it online or in other published work.

The lacZ insertion in *opa* gene (called 3-66) is described in section 7.4 of Ruth Beckervordersandforth's thesis (see the link below)
https://publications.uni-mainz.de/theses/frontdoor.php?source_opus=1536

2- The expression pattern of the line outside of glia seems to be unpublished and the authors need to provide evidence that "opa-lacZ" actually reports the pattern of opa expression. Immunolabeling with anti-opa antibodies or comparison with the expression patterns of other lines (such as the opa-lacZ line published by Cimborá & Sakonju in 1995) are possible approaches.

So far, the only anti Opa antibody available is described in Benedyk and al., 1994. We used this antibody and unfortunately it does not work at all. The same observation has been already mentioned in Clark and Akam, 2016.

During our work, we generated an antibody against Opa. We were only able to detect Opa protein with our antibody when *opa* is ectopically expressed; but unfortunately, we cannot detect the endogenous Opa protein.

We also had troubles detecting *opa mRNA* by in-situ hybridization in the VNC. In addition, we have used the *opa-lacZ* line published by Cimborá & Sakonju, (1995), but it does not fully recapitulate all the *opa* expression pattern.

3- The authors restrict their attention to CCAP and Burs neurons of the VNC, but why is not obvious. Confusingly, they also state (l. 180) that expression of Burs is restricted to T3-A10 in the VNC, but the paper they cite describes expression also in the subesophageal ganglion. Why are these neurons not mentioned, and why did the authors restrict their analysis to CCAP and Burs neurons in the VNC and not include the neurons of the SEZ and central brain. Also, in Fig.s 1 and 2A, the CCAP neurons of the central brain don't even seem to be labeled by opa-lacZ, which runs counter to the manuscripts conclusions. Are these neurons an exception, or is opa-lacZ labeling just very dim in them?

We have written "In control adult VNC..." instead of "In control adult CNS ..." (l. 180 before changes) given that we were talking about Burs expression in the VNC.

Although we observed opa-lacZ expression in all the CCAP neurons of the CNS (including the ones in the central brain) we state in the manuscript that for technical reason we focus our attention on CCAP neurons in the VNC as others authors have also done (Veverytsa and Douglas W. Allan, 2011, 2012; Lee, Kikuno et al., 2013)

4- The experiments analyzing the effects of opa knockdown on the survival of CCAP and Burs neurons are somewhat confusing and inconclusive (see l. 147 and Fig.s 3&4). This is because the only assessment of survival is GFP expression or Burs immunolabeling (i.e. there is no direct assay for cell death). It could thus be that RNAi-opa reduces CCAP and Burs gene expression—sometimes below the signal detection threshold, but without killing the cells—and that UAS-p35 completely restores the levels of CCAP, but not Burs expression. Evaluation of the results is also hampered by the fact that there is no explicit control demonstrating that UAS-p35 is inhibiting apoptosis. Since the latter reagent is novel, a positive control demonstrating its efficacy is really necessary, although even such a demonstration would not rule out the possibility that it is merely regulating peptide expression (rather than apoptosis) in the case of the CCAP/Burs neurons.

We do not agree that UAS-p35 is a novel reagent and its efficiency to block specifically apoptosis has long been demonstrated (Hay et al, 1994). It is a tool commonly used to block apoptosis in the CNS (Hara et al, 2018; Smart et al, 2017; Lovick et al, 2016) or in other tissue (Nakazawa et al., 2016; Guo et al, 2014). Also Lee and al. (2013) have used UAS-p35 to prevent CCAP neurons from apoptosis.

The rescue of Bursicon cell number in CCAP>RNAi opa vs CCAP>RNAi opa UAS p35 was also obtained by blocking apoptosis expressing UAS-RHG.miRNA instead of UASp35 (we have introduced this data in the new version of the manuscript). Therefore, these results clearly indicate that downregulation of opa in CCAP neurons lead to cell death. We do not consider necessary to perform a direct assay for cell death.

5- Although the fidelity of expression of the CCAP-Gal4 line has been well established in the CNS by immunostaining, it's fidelity outside of the CNS has not. While it is extremely likely that the novel CCAP-Gal4 cells described in the manuscript do in fact express CCAP, it would desirable if this were either directly confirmed by immunostaining or the conclusions of the paper on this point tempered.

We thank the referee for this comment. We do not have the CCAP antibody to test whether the novel CCAP-Gal4 cells express CCAP, however staining with Bursicon and MIP reveals that all the CCAP-Gal4 cells express both neuropeptides, and therefore we conclude that these neurons are indeed CCAP neurons. We have added Burs staining in Fig7. We also send the referee MIP staining in a CCAP neuron innervating the oviduct.

6- l. 93-4 *The evidence that the late differentiating CCAP neurons are necessary and sufficient for pupal ecdysis, as originally reported, is inconsistent with results reported more recently by Diao et al. (Genetics, 2016). It would be useful to mention both sets of results.*

We thank the referee for pointing out this omission. The reference has been added.

7- l. 125 *“miss position” What is meant here is unclear. Also, an arrow in the figure panel pointing to the bristle phenotype would be useful.*

We added arrows in Fig. 3A, B and C to point out the postscutellar bristles and add the corresponding changes in the figure legend. The phrase has been changed.

8- l. 136 *“are composed of” should read “include”*

We have changed “are composed of” for “include”.

9- l. 143 *The decrease (13->4) looks to be 3-fold not 4-fold*

We thank the referee for pointing out this mistake. We have changed “4-fold” for “3-fold”:

10- l. 144-5 *Here and elsewhere the authors make an assertion about behavior (i.e. “alteration in the execution of the postecdysis sequence”) when all they appear to have examined is the endpoint (expansion of the wings). Since expansion failure can have causes other than behavioral deficits, and successful expansion can occur even in the absence of normal behavior. Some caution should therefore be exercised in the wording to indicate only that the end phenotype is either different or normal.*

Referee 2 made a similar comment (see our response to comments 3 and 4 of Referee 2)

*11- l. 159 “the quantified number of CCAP cells” presumably Burs cells are meant since this is what is reported in Fig. 4, but it would be interesting to know if here and in Fig. 3 the numbers of neurons that express CCAP, but NOT Burs is also reduced by RNAi-*opa* expression.*

The number of neurons that expressed CCAP but not Burs are also reduced by RNAi-*opa* expression.

12- l. 178 What is meant by a “mild” phenotype?

Flies with a mild phenotype present wings partially expanded and thorax partially extended.

*13- l. 212 The ability of *opa* to activate Burs expression is actually rather limited based on the data presented and the conclusion that it does so “in many but not all VNC cells” seems exaggerated.*

We have rewritten this conclusion as “Taken all these results together, we conclude that *Opa* is able to activate Burs in VNC cells.”

14- l. 223 and Fig. 7E: the text indicates that only one neuron is labeled, but the amount of labeling in the figure makes it look like more. Is it really only a single neuron?

Indeed only one neuron is labeled. The strong GFP signal observed in the lumen of the ejaculatory bulb indicates that this organ is highly innervated by this neuron. Also, as we mention in the text, the previous Fig7E (now referred to as Fig7F) shows a projection of stacks of the whole ejaculatory bulb, what does not allow to see well the unique CCAP neuron. You will find below a Z stack of the projection of the ejaculatory bulb (click on the photo to see the Z stack).

FL

opa-lacZ

DAPI

CCAP>GFP

15- Fig. 2B mentions data from a line called “386-Gal4” and several neurons are said to express myomodulin based on a paper not cited in the references. The presumed paper, however, does not refer to the stated Gal4 line and the manuscript/figure provides no independent evidence that myomodulin is expressed in the indicated neurons. Also, the 386-Gal4 line is not listed in the Materials and Methods though a 386Y-Gal4 line is—citing however a different paper. From that paper, the connection between 386Y-Gal4 and myomodulin is also not apparent.

In the legend of Fig. 2B, we had made a mistake. The driver is 386Y-Gal4. This mistake has been corrected.

In O’Brien and Taghert, 1998, it is shown that the 36Y-Gal4 driver is expressed in the myomodulin cells located in the VNC (called VAs cells in the paper). These cells are easily identifiable by their number and position in the VNC. In Taghert et al., 2001, where the 386Y-Gal4 line is described, it is mentioned that the expression pattern of 386Y-Gal4 includes that of 36Y-Gal4. Therefore, the VAs cells also expressed the 386Y-Gal4. We have clarified this confusion in the legend of Fig2B.

16- Fig. 3B’: what criteria were used to select the eL3, mL3, and IL3 animals?

We use larval size as criterium in a synchronized culture.

Referee: 2

1- The images that show co-localization of CCAP>GFP + opa-LacZ (Figs 1, 2), CCAP>GFP + dac + bursicon-IR (Figs. 3, 4, 5) are too low resolution and are maximal projections. The authors need to include higher magnification, single plane, images of at least some of the neurons to document co-localization; this is especially true for adult preparations, where neurons occur in a “clump”. Also, the expression of dac (Figs. 3, 4, 5) is barely visible (and it isn’t really clear why dac expression is included). If the authors wish to include this gene, these images in particular need to be greatly improved.

As the referee requested, we have added single confocal planes of CCAP neurons showing CCAP>GFP and opa-lacZ colocalization in Fig2.

Dac is expressed in CCAP motor neurons but not in CCAP interneurons. In Figs. 3, 4 and 5 we stained against Dac to see whether opa misexpression could alter one CCAP neuron type but not the other, which it has not been the case. In the new version of the manuscript we have added a panel in grey for better visualization Dac expression in Fig 3,4 and 5. We have also included a zoom of the CCAP motoneuron and CCAP interneuron in Fig 5 to better show that ectopic expression of Opa or Zic2 in the CCAP interneuron leads to ectopic expression of Burs but these neurons still remain Dac negative.

2- The images that illustrate the post-emergence defects of the adults are not good enough quality. This is especially true of Figs. 3A-C, where a good part of the fly is out of focus. The corresponding images in Figs. 5A-C are better but not great. The age of the flies (number of hours after emergence) should be indicated in the legends for Figs 3A-C, 4A,B and 5A-C.

Indeed, parts of the fly, concretely heads and abdomens, are out of focus in Fig3A-C because of technical limitations. However, we do not think that this is relevant. What is important is to see the thorax (melanisation, sclerotisation and the bristles), which is in focus.

The age of the flies Figs 3A-C, 4A,B and 5A-C is indicated in the legends.

3- The title and the text state that opa controls “adult postecdysis behavior”, yet no evidence is presented that behavior itself is affected. The fact that wing inflation is defective could be due to defects in the eversion of the wing imaginal disc at pupation (see 4 below). In addition, cuticle pigmentation is not a behavior. So, unless the authors specifically looked at postecdysis behaviors (e.g., wing inflation behaviors) and report on them, I would re-word these statements saying that “opa plays a role in post-emergence maturation” or something to that effect.

The fact that no defects are observed in pupa (see answer to 4) suggests that indeed opa controls adult postecdysis behavior.

4- *The postemergence wing expansion defects reported could be caused by defects at pupation, as incorrectly everted wing discs at pupal ecdysis will also lead to defects in the adult wing expansion. Thus, the authors should examine whether the animals pupate correctly. This can easily be determined by looking at pharate adults. Defects at pupation cause wings and legs to be shorter than normal (and, in extreme cases, for the head to be incorrectly everted), due to defective head and imaginal disc eversion.*

We have carefully analyzed the pharate adults. These animals pupate correctly: wing and legs are perfectly patterned and as long as in normal flies and heads evert properly. We have introduced a sentence in the manuscript addressing this point.

5- *Results lines 127-135. Although the results do suggest that opa is required continuously for the expression of a normal post-emergence phenotypes, Figure 3 indicates that there might be a sensitive period between the 3rd instar and the early pupa, suggesting a possible role in pupal ecdysis (see point 4, above). It would be useful if the authors did the complementary experiment, of raising the temperature in a staggered series starting in the larva (and/or restricting it to the period around pupation).*

We have done the suggested experiments and got similar results: the earlier opa expression is down regulated, the highest is the probability to have an adult fly with post-eclosion defect.

6- *CCAP neurons are marked by using reporter line. Thus, CCAP expression itself is not measured, but GFP. This needs to be corrected throughout the text, changing "CCAP expression" for "CCAP reporter expression", or something to that effect.*

We performed those changes.

7- *Line 93-94: the findings of Veverytsa and Allan (2012) have recently been put into question (Diao et al. (2016) Genetics, Vol. 202, 175-189)*

We added this reference,

8- *The experiment shown in Figure 4 should really use the genotypes: CCAP>opa RNAi + dcr2 vs CCAP>opa RNAi + dcr2 + p35, in other words showing what happens when p35 is co-expressed with opa RNAi in CCAP neurons vs when opa RNAi is expressed alone (with dcr2) in CCAP neurons.*

What we wanted to study here was to analyze the effect of opa downregulation when cell death was blocked. Therefore, the proper control experiment is by expressing UAS p35.

9- *The stippling in Figs 3F, 4E, 4F and 5G is distracting and seems unnecessary; white, grey and black would be sufficient.*

We decided not to follow the suggestion of the referee.

10- *The English needs to be improved throughout the text. In particular:*

+ title: "Implication ..." should be "Involvement ..."; although "Role of..." might be preferable/ We chose to write "Role of..." instead of "implication..."

+ line 62: not clear why "thus" is used. Sentences in line 62-65 are unclear.

+ lines 144-146 are a little unclear.

+ line 173, should say "folded wings"

+ legend to Fig. 3 should read "Knockdown of opa leads to ..." (vs "Lack of opa leads to ...").

+ legend to Fig. 5 should read "Overexpressing opa..." (vs "Overexpression opa...").

We thank the referee for pointing out all these mistakes. We have corrected them.

Appendix B

Reviewer(s)' Comments to Author:

Referee: 2

The authors have addressed most of my concerns. I would just insist that they not refer to the adults defects as being in "postecdysis behavior" since a) they did not examine the fly's behavior, only the terminal phenotype (unexpanded wings) b) there seems to be additional non behavioral post-eclosion defects including in cuticle hardening (and possibly also melanization; difficult to tell from the pictures).

Referee: 1

The authors have addressed most of my concerns. I would ask only that they make the following changes: l76 and elsewhere: change "behavior" to "maturation" as suggested by Reviewer 2. (Failure of flies to secrete bursicon into the hemolymph can give rise to all the phenotypes observed without necessarily affecting postecdysis behavior, so it is best not to explicitly attribute the deficits to behavioral failure in the absence of further data.)

l355-6: Replace "its localization is described..." with "the location of the p-element insertion is described..."

l235-6: Say: "Further studies are required to confirm the presence of CCAP and to confirm the role of this neuropeptide in these cells."

Discussion, Section 1: Include a statement that you have been unable to directly confirm by either immunostaining or in situ labeling that the expression seen with opa-LacZ faithfully mimics the expression of the gene. (This can be coupled to a statement about why you think the labeling pattern is nevertheless faithful.)

Following the comments of Referee 1 and 2 we performed the following changes:

- We substituted the word behavior for maturation (in the title and throughout the text).

- We replaced in l355 "its localization is described..." with "the location of the p-element insertion is described..."

- We wrote in l235 "Further studies are required to understand the role of the neuropeptide Burs in these cells." Here we are referring to the neuropeptide Burs but not the neuropeptide CCAP as Referee1 underlined,

- In Discussion, Section 1, we changed "opa expression" for "opa-lacZ expression" for better clarification.